# A Similarity-based Approach for Efficient Large Quasi-clique Detection

## ABSTRACT

Identifying dense subgraphs called quasi-cliques is pivotal in various graph mining tasks across domains like biology, social networks, and e-commerce. However, recent algorithms still suffer from efficiency issues when mining large quasi-cliques in massive and complex graphs. Our key insight is that vertices within a quasi-clique exhibit similar neighborhoods to some extent. Based on this, we introduce NBSim and FastNBSim, efficient algorithms that find near-maximum quasi-cliques by exploiting vertex neighborhood similarity. FastNBSim further uses MinHash approximations to reduce the time complexity for similarity computation. Empirical evaluation on 10 real-world graphs shows that our algorithms deliver up to three orders of magnitude speedup versus the state-of-the-art algorithms, while ensuring high-quality quasi-clique extraction.

## CCS CONCEPTS

• **Theory of computation → Graph algorithms analysis**; • **Mathematics of computing → Graph algorithms**.

## KEYWORDS

Quasi-cliques, neighborhoods, similarity, MinHash

**ACM Reference Format:**
Anonymous Author(s). 2018. A Similarity-based Approach for Efficient Large Quasi-clique Detection. In *Proceedings of Make sure to enter the correct conference title from your rights confirmation emai (Conference acronym 'XX)*. ACM, New York, NY, USA, 9 pages. https://doi.org/XXXXXXX.XXXXXXX

## 1 INTRODUCTION

Dense subgraph extraction from large graphs has emerged as a key operation in graph mining. By identifying highly interconnected groups of vertices, dense subgraphs enable the discovery of critical components hidden in real-world networks. For example, dense subgraph mining has been used to identify spam link farms in web graphs [13, 31], discover regulatory motifs in genomic DNA [12], compress graphs [3], and analyze social network [15, 38]. The widespread utility of dense subgraphs underscores their importance as a fundamental graph mining primitive.

Various formulations for extracting different classes of dense subgraphs have been proposed based on different density metrics. Clique is the most classic dense subgraph model, which requires full connectivity between all vertices. While conceptually simple,

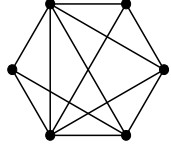

(a) A clique     (b) A quasi-clique with $\alpha = 0.8$

**Figure 1: Illustrating clique and quasi-clique.**

cliques are often unrealistic for noisy, incomplete real-world data. This has motivated the development of various flexible dense subgraph formulations in the literature.

Quasi-cliques represent a relaxation of the clique concept to allow for real-world noise and missing edges. An $\alpha$-quasi-clique is a subgraph where the number of edges is at least $\alpha$ times the number of edges in a clique of the same size, for some density parameter $\alpha \in (0, 1)$. Figure 1 illustrates a clique, which is fully connected, versus a quasi-clique which misses some edges. In the quasi-clique shown, there are 15 possible edges but only 12 are present, giving an edge-density of $12/15 = 0.8$. Quasi-cliques provide a more flexible formulation by only requiring the subgraph to be nearly fully connected based on the edge-density threshold $\alpha$. This makes them better suited for real-world graphs compared to strict cliques.

**Prior work.** The maximum quasi-clique problem (MQCP) aims to find the largest $\alpha$-quasi-clique in a graph but is NP-hard to compute [24]. Thus, several heuristic algorithms have been proposed. Konar and Sidiropoulos [16] presented a polynomial-time algorithm NB that mines large quasi-cliques from vertex neighborhoods based on clustering coefficients. NB achieves the state-of-the-art performance versus prior methods [22, 35] by refining well-chosen neighborhoods.

However, as NB treats entire neighborhoods as quasi-cliques, it risks overlooking dense subgraphs contained within larger neighborhoods. Consider a vertex $v$ where the neighborhood $N(v)$ has size 110 and edge-density 0.5. While $N(v)$ may not be optimal as a whole, it could contain a dense subset $S$ of size 70 with edge-density 0.9 that meets the quasi-clique threshold, but $S$ is *missed by* NB.

Furthermore, NB has a high complexity of $O(m^{3/2})$ as it computes the clustering coefficient for every vertex's 1-hop neighbors via triangle counting, including irrelevant ones. In summary, NB has limitations in both accuracy and efficiency.

**Our solution.** To address the limitations of prior quasi-clique extraction methods like NB, we propose NBSim and FastNBSim, novel quasi-clique extraction algorithms based on similarity measures.

Different from NB, when checking each neighborhood (e.g., $u$'s neighborhood), NBSim treats each neighbor $v$ in the neighborhood as a unit, instead of treating the entire neighborhood as a whole. Specifically, NBSim decides whether to include each vertex $v$ from

$u$'s neighborhood based on the similarity between the neighborhoods of $u$ and $v$, and has the ability to detect the dense quasi-clique inside the whole neighborhood of $u$. This enables the extraction of quasi-cliques missed by NB that are solely based on clustering coefficients of whole neighborhoods. Theoretically, we prove that this similarity-based algorithm provides a lower bound on edge-density for extracted quasi-cliques. Besides, it avoids expensive clustering coefficient computations as NB needs. We have also devised a new vertex ordering strategy to further enhance efficiency. By pruning unpromising neighborhoods early, we significantly reduce unnecessary computations on irrelevant vertices.

To further speed up the similarity computation, we propose an algorithm FastNBSim that uses the MinHash technique to estimate the similarity between two neighborhoods in constant time, rather than the neighborhood size-dependent time of the exact computation. Experiments show that even using small-size signatures of MinHash, we can achieve promising results compared to NBSim.

Our principal contributions are summarized as follows:

- A similarity-based algorithm NBSim can efficiently detect quasi-cliques inside the neighborhoods.
- A novel vertex ordering strategy prioritizes vertices whose neighborhoods have a high potential to contain large quasi-cliques first.
- A MinHash-based algorithm FastNBSim incorporates approximations with MinHash to reduce the overall time complexity to linear.
- Comprehensive experiments on 10 real-world datasets demonstrate that our solutions, especially FastNBSim, are up to three orders of magnitude faster than state-of-the-art baselines while ensuring high-quality quasi-cliques extraction.

**Outline.** The rest of the paper is organized as follows. We review the related work in Section 2. Section 3 gives the preliminaries and the definition of the maximum quasi-clique problem. We present our similarity-based algorithm NBSim in Section 4 and MinHash-based fast algorithm FastNBSim in Section 5. Experimental results are presented in Section 6. We conclude the paper in Section 7.

## 2 RELATED WORK

Finding dense subgraphs in large graphs is an important task in graph mining. Among different dense subgraph models, clique is the archetypal one. Here, we review related work on cliques and some close variants, e.g., the densest subgraph and quasi-cliques.

**Maximum Clique Finding.** The maximum clique problem aims to find a clique of maximum size in a given graph. This problem is NP-hard. Branch-and-bound search methods have been extensively studied for finding maximum cliques exactly [4, 5, 18, 23, 30, 39]. The key idea is to grow an initially empty clique by recursively moving vertices from a candidate set to the clique, pruning branches that cannot lead to a maximum clique. For sparse graphs, Chang [6] proposes more efficient maximum clique finding algorithms.

**Maximal Clique Enumeration.** The problem of maximal cliques enumeration (MCE) in a graph is harder than MCC, since the output size may itself be very large. MCE has also been extensively studied. Bron and Kerbosch [2] introduced a backtracking search method to enumerate maximal cliques. Tomita et al. [32] used the idea of "pivoting" in the backtracking search. Eppstein et al. [11]

used a degeneracy-based vertex ordering schema on top of the pivot selection strategy. In [8], Das et al. present shared-memory parallel algorithms for MCE.

**Densest Subgraph Discovery.** Due to the NP-hardness of clique finding, a less stringent, polynomial-time formulation for mining dense subgraphs is formulated. Goldberg [13] introduced the densest subgraph (DS) discovery problem and proposed the maximum-flow-based algorithm to seek the subgraph with maximum density, i.e. ratio of edges to vertices. Variants of this problem include generalizing the density measure to $k$-clique-based density [34], finding dense subgraphs in bipartite graphs [22] and directed graphs [19], and finding dense subgraphs in evolving graphs [10].

**Maximum Quasi-Clique Problem.** Another relaxation to cliques is known as quasi-cliques. Given a threshold $\gamma$, a $\gamma$-quasi-clique is a subgraph with edge density above $\gamma$. The maximum quasi-clique problem aims to find the largest $\gamma$-quasi-clique in a graph. This problem generalizes maximum clique finding and is NP-hard [24].

Algorithms for this problem can be classified as exact or heuristic. Exact algorithms such as branch-and-bound can guarantee optimality but have high runtime on large graphs [20, 21, 24, 27, 37]. Thus, many efficient heuristic algorithms have been developed. Abello et al. [1] introduced an efficient semi-external memory algorithm and relies on greedy randomized adaptive search procedures. Tsourakakis et al. [35] studied the optimal-quasi-clique and designed a greedy algorithm and a local-search algorithm for MQCP. Pinto et al. [25, 26] proposed the biased random-key genetic algorithm for the MQCP. Djeddi et al. [9] used an extension of adaptive multi-start tabu search to approximate the MQCP solution. Konar and Sidiropoulos [16] proposed an efficient algorithm for MQCP by mining large quasi-cliques from vertex neighborhoods. Recently, Chen et al. [7] developed an efficient local search algorithm by taking a novel vertex selection strategy. However, state-of-the-art heuristics remain inefficient on massive graphs. The algorithm in [7] provides no polynomial time guarantees, while the polynomial method in [16] has high runtimes on large graphs.

## 3 PRELIMINARIES

### 3.1 Problem Definition

We consider an unweighted and undirected graph $G = (V, E)$, where $V$ and $E$ are the sets of vertices and edges respectively. We denote the numbers of vertices and edges in $G$ by $n$ and $m$ respectively. For a vertex $u$, the neighborhood $N(u)$ consists of the set of nodes that are neighbors of node $u$ and $u$ itself [41]. The degree of $u$ is defined as the number of neighbors of $u$, denoted as $d(u)$, i.e., $|N(u)| = d(u) + 1$. Given a subset of vertices $S \subseteq V$, denote $E(S)$ as the subset of $E$ containing edges only between the vertices in $S$, i.e., $E(S) = E \cap (S \times S)$. We use $G[S] = (S, E(S))$ to denote the subgraph induced by $S$, and $d_S(u)$ to denote degree of $u$ in $G[S]$.

*Definition 3.1 (Edge-density [7, 16]).* Given a graph $G = (V, E)$ and its subgraph $G_S = (S, E(S))$ induced by $S$, its edge-density is defined as:

$$\delta(S) = \frac{|E(S)|}{\binom{|S|}{2}} \tag{1}$$

A clique is a subset of vertices such that every two distinct vertices in the clique are adjacent, i.e., $\delta(S) = 1$ when $S$ is a clique.

Given a parameter $\alpha \in (0, 1]$, a subgraph $G[S]$ is said to be a $\alpha$-quasi-clique if $\delta(S) \geq \alpha$, i.e., if the number of its edges is at least $\alpha \cdot \binom{|S|}{2}$.

*Definition 3.2 (Maximum quasi-clique problem (MQCP)).* Given a graph $G = (V, E)$ and density threshold $\alpha \in (0, 1]$, the maximum quasi-clique is an $\alpha$-quasi-clique $S \subseteq V$ with maximum cardinality.

MQCP is proved to be an NP-hard problem [24]. Hence, we will start by presenting an algorithm called `NBSim` that can find approximate maximum quasi-cliques in polynomial time. We will then improve this algorithm to a faster, linear time version called `FastNBSim`, using minhash approximation.

Before we present our similarity-based algorithms, we first give the definition of structure similarity. Given two neighboring vertices $u$ and $v$, the similarity $\sigma(u, v)$ between $u$ and $v$ is defined as the set similarity between $N(u)$ and $N(v)$. In existing studies [29, 41], Jaccard similarity is adopted to measure the similarity. The definition of Jaccard similarity is as follows:

*Definition 3.3 (Jaccard similarity [14]).* Given two sets $X$ and $Y$, the Jaccard similarity between these two sets is defined as $\frac{|X \cap Y|}{|X \cup Y|}$.

Based on Jaccard similarity, the similarity between two vertices $u$ and $v$ is defined as $\sigma(u, v) = \frac{|N(u) \cap N(v)|}{|N(u) \cup N(v)|}$.

## 4  NBSIM: A SIMILARITY-BASED ALGORITHM

In this section, we develop a novel polynomial-time algorithm for finding near-maximum quasi-cliques. Our algorithm exploits the existence of dense vertex neighborhoods of non-trivial sizes in real-world graphs, as proven in Theorem 3.5 of [16]. This theorem demonstrates that large, high-quality quasi-cliques can be extracted from neighborhoods with sufficiently high edge density. Our algorithm has two key components. We first introduce our similarity-based vertex selection strategy to extract high-quality quasi-cliques from neighborhoods. We then speed up this algorithm further by incorporating pruning techniques based on ordering and bounds.

### 4.1  Similarity-based Vertex Selection

In this subsection, we focus on extracting quasi-cliques from the neighborhood of a single vertex. As the vertices within a quasi-clique tend to exhibit a higher level of similarity between each other compared to vertices outside the quasi-clique, a straightforward idea to extract quasi-cliques from the neighborhood of $u$ is to find neighbors with high similarities to $u$. However, the Jaccard similarity is not very suitable for our case, while it was used in many applications, such as graph clustering [33, 40, 41]. To find large quasi-cliques from the neighborhood of $u$, our goal is to find vertices $\{v | v \in N(u)\}$ such that $N(u)$ is mostly contained by $N(v)$, instead of finding a vertex v with $N(v)$ similar to $N(u)$. Hence, we propose a new metric, containment score, as the vertex selection criterion.

*Definition 4.1 (Containment score).* Given two vertices $u$ and $v$, the containment score of $u$ in $v$ is defined as

$$t(u, v) = \frac{|N(u) \cap N(v)|}{|N(u)|} \qquad (2)$$

---

**Algorithm 1:** Extract quasi-clique from $N(u)$: QCextract

**Input:** vertex $u$, threshold $\gamma \in (0, 1]$, $b \in (0, 1]$
**Output:** a vertex set extracted from $N(u)$

1  $S \leftarrow \emptyset$
2  **for** *each vertex $v$ in $N(u)$* **do**
3  $\quad$ **if** $t(u, v) \geq \gamma$ **then** $S \leftarrow S \cup \{v\}$
4  **if** $\frac{|S|-1}{|N(u)|} < b$ **then** $S \leftarrow \emptyset$
5  **return** $S$

---

Compared to Jaccard similarity, the containment score is an asymmetric definition, i.e., $t(u, v)$ might not equal to $t(v, u)$, which allows the scenario that only part of $N(v)$ highly overlaps with $N(u)$. Assuming that $N(u)$ induces the ideal quasi-clique, all vertices in $N(u)$ can be selected based on the containment score, which might not be achieved via Jaccard similarity. We further use the following example to illustrate the advantage of containment score.

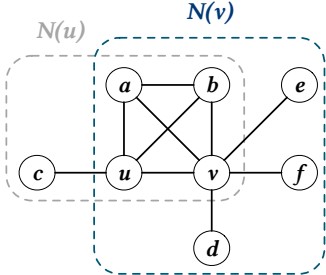

**Figure 2: Subgraph induced by $N(u)$ and $N(v)$.**

*Example 4.2.* Figure 2 depicts a subgraph induced by $N(u)$ and $N(v)$. The neighborhoods of $u$ and $v$ are denoted as $N(u) = \{a, b, c, v, u\}$ and $N(v) = \{a, b, d, e, f, u, v\}$, respectively. Suppose we want to extract a 0.8-quasi-clique from $N(u)$. Using Jaccard similarity, $\sigma(u, v) = \frac{|N(u) \cap N(v)|}{|N(u) \cup N(v)|} = \frac{4}{8} = 0.5$. Based on this, we may exclude $v$ from the quasi-clique. However, the containment score gives $t(u, v) = \frac{|N(u) \cap N(v)|}{|N(u)|} = \frac{4}{5} = 0.8$, suggesting $v$ could be included. Importantly, $a, b, u, v$ forms a clique in the subgraph. This validates the appropriateness of including $v$ based on the containment score.

Based on the above discussion, we present the algorithm to extract quasi-cliques from neighborhoods, named QCextract, in Algorithm 1. QCextract takes a vertex $u$ and threshold values $\gamma$ and $b$ as input. It aims to extract a vertex set $S$ from $N(u)$, where the edge-density of $G[S]$ is controlled by $\gamma$ and $b$. It first initializes an empty set $S$ to store the result (line 1). Then it iterates over each vertex $v$ in the neighborhood $N(u)$ of vertex $u$ (line 2), checks whether the containment score $t(u, v)$ exceeds the threshold $\gamma$, and adds vertex $v$ to the set $S$ if the condition is fulfilled (line 3). If the extracted set $S$ is smaller than $b|N(u)|$, which implies only few vertices of $N(u)$ have highly overlapped neighbors with $u$, we will set $S$ to $\emptyset$ (line 4). Finally, QCextract returns the extracted set $S$ (line 5).

If $S$ is not empty, we show that the lower bound of the edge-density of $S$ is determined by $\gamma$ and $b$, as follows:

THEOREM 4.3 (LOWER BOUND OF EDGE-DENSITY OF QUASI-CLIQUE RETURNED BY ALGORITHM 1). *Given a graph $G$, a vertex $u$, and threshold parameters $\gamma$ and $b$, the edge-density of the quasi-clique $S$ returned by Algorithm 1 is lower bounded by, if $S$ is not empty:*

$$\delta(S) \geq 1 - \frac{1-\gamma}{b} \tag{3}$$

PROOF. According to line 3 of Algorithm 1, each vertex $v$ in $S$ satisfies that $t(u,v) \geq \gamma$, which implies that $|N(u) \cap N(v)| \geq \gamma|N(u)|$. According to line 4 of Algorithm 1, we have $|S| - 1 \geq b|N(u)|$.

For a specific $v \in S$, denote $t(u,v) = \gamma'$ and $|S| - 1 = b'|N(u)|$. By defining $S' = S \setminus v$, we have $|S'| = b'|N(u)|$.

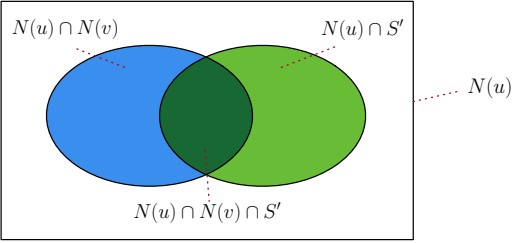

**Figure 3: Illustrating the lower bound of $d_S(v)$.**

By applying the inclusion-exclusion principle (ref. Figure 3), we can infer that the degree of $v$ in $G[S]$ satisfies

$$d_S(v) = |N(u) \cap N(v) \cap S'| \geq (b' - (1-\gamma'))|N(u)| \tag{4}$$

Hence, we obtain

$$\frac{|N(u) \cap N(v) \cap S'|}{|S'|} \geq \left(1 - \frac{1-\gamma'}{b'}\right) \tag{5}$$

$$\geq \left(1 - \frac{1-\gamma}{b}\right), \tag{6}$$

which means that $\forall v \in S, d_S(v) \geq \left(1 - \frac{1-\gamma}{b}\right)(|S| - 1)$. Hence, the edge-density $\delta(S)$ is at least $1 - \frac{1-\gamma}{b}$, as $|E(S)| = \frac{\sum_{v \in S} d_S(v)}{2}$. □

**Remark.** The proof shows that each vertex in $G[S]$ is connected to at least $\left(1 - \frac{1-\gamma}{b}\right)(|S| - 1)$ other vertices within $G[S]$, which is a stricter requirement than Definition 3.1. Definition 3.1 only requires that the overall density of the subgraph $G[S]$ is larger than a specific threshold value.

**Effect of parameters.** By increasing both $\gamma$ and $b$, the term $\frac{1-\gamma}{b}$ approaches 0. Consequently, the value of $\delta(S)$ approaches 1, and Algorithm 1 tends to output near-cliques. Although high $\gamma$ and $b$ values may cause Algorithm 1 to return an empty set for the neighborhoods of some specific nodes, we still have a high probability of finding the large quasi-cliques from the whole graph, because dense vertex neighborhoods of non-trivial sizes exist in real-world graphs, according to Theorem 3.5 of [16].

To obtain an $\alpha$-quasi-clique as in Lemma 4.3, we must set $b$ and $\gamma$ such that $1 - \frac{1-\gamma}{b} > \alpha$. In particular, $b$ controls the similarity requirement between $S$ and $N(u)$. Setting $b$ close to 1 makes Algorithm 1 return almost the entire neighborhood itself.

---

**Algorithm 2:** Find near-maximum quasi-clique: `NBSim`

**Input:** Graph $G$, threshold $\gamma \in (0, 1]$, $b \in (0, 1]$
**Output:** A near-maximum quasi-clique

1   $S \leftarrow \emptyset$
2   **for** *each vertex $u$ in descending $\gamma$-degree order* **do**
3      **if** $d_\gamma(u) < |S|$ **then** break
4      $C \leftarrow \text{QCextract}(u, \gamma, b)$
5      **if** $|C| > |S|$ **then** $S \leftarrow C$
6   **return** $S$

---

## 4.2 Pruning via Ordering and Bound

In Section 4.1, we presented an algorithm to extract quasi-cliques from a single vertex's neighborhood. To find near-maximum quasi-cliques across the full graph, an exhaustive approach is to extract from every neighborhood. However, this involves significant unnecessary computation. To improve efficiency, we first derive the size upper bound of quasi-cliques extractable from each neighborhood. We then propose a vertex ordering strategy to prune unpromising neighborhoods.

We first give a simple upper bound based on degree for Algorithm 1, which is also an upper bound for maximum clique computation [28].

LEMMA 4.4 (DEGREE-BASED UPPER BOUND). *For a graph $G$ and a vertex $u$ in $G$, the size of the set returned by $\text{QCextract}(u)$ is no larger than $d(u) + 1$.*

The lemma follows from that all vertices returned by $\text{QCextract}(u)$ are from the neighborhood of $u$. However, this upper bound is quite loose.

Inspired by core numbers from $k$-core, a subgraph model where each vertex has at least $k$ neighbors within the subgraph, we propose a new concept $\gamma$-degree, which is a tighter upper-bound for the returned quasi-clique.

*Definition 4.5 ($\gamma$-degree).* Given a graph $G$ and a vertex $u$, we define the $\gamma$-degree of $u$ as the number of neighbors of $u$ with a degree at least $\gamma * d(u)$, denote as $d_\gamma(u)$.

$$d_\gamma(u) = |\{v \in N(u) \mid |N(v)| \geq \gamma \cdot |N(u)|\}| \tag{7}$$

Note that $u$ is counted in its $\gamma$-degree $d_\gamma(u)$ but not in its original degree $d(u)$.

LEMMA 4.6 ($\gamma$-DEGREE-BASED UPPER BOUND). *For a graph $G$ and a vertex $u$ in $G$, the size of the set returned by $\text{QCextract}(u)$ is no larger than $d_\gamma(u)$.*

PROOF. In Algorithm 1, for a vertex $v$ to fulfill the condition $t(u,v) > \gamma$, $|N(v)|$ must be at least $\gamma \cdot |N(u)|$. Hence, the size of the returned set is upper bounded by $d_\gamma(u)$ via Definition 4.5. □

As $d_\gamma(u) \leq d(u) + 1$ holds for every $u \in V$, the $\gamma$ degree-based upper bound is tighter than the degree-based upper bound. Besides, the $\gamma$-degree for every vertex in $G$ can be efficiently computed by iterating over each vertex $v \in N(u)$ to check whether $|N(v)| > \gamma \cdot |N(u)|$ fulfills, in $O(m)$ total time.

Based on the above discussions, we propose the algorithm `NBSim` for computing near-maximum quasi-clique. The pseudocode is

shown in Algorithm 2. The algorithm initializes an empty set $S$ (line 1). Then, it iterates through each vertex $u$ in descending order with respect to the $\gamma$-degree (line 2). It compares the $\gamma$-degree of vertex $u$ with the size of $S$. If the $\gamma$-degree is less than the size of $S$, the loop breaks, as there is no possibility of forming a larger quasi-clique via Lemma 4.6 (line 3). For each vertex $u$ satisfying the degree condition, the algorithm proceeds to construct a vertex set $C$ by invoking QCextract($u, r, b$) (line 4). If the size of set $C$ is greater than the size of $S$, $S$ is updated to $C$ (line 5). Finally, $S$ is returned as the near-maximum quasi-clique.

**Complexity**. Let $d_{max}$ denote the maximum degree of any vertex in the graph. The time complexity of Algorithm 2 is $O(m \cdot d_{max})$ Because it calls QCextract for each vertex $u$, and QCextract($u$) will compute $t(u, v)$ for $|N(u)|$ times. In total, we need to compute $t(u, v)$ for each edge twice, and the cost to compute $t(u, v)$ is $O(d_{max})$. Hence, the overall time complexity is $O(m \cdot d_{max})$

In NBSim, we need to choose two user-defined parameters $\gamma$ and $b$, which will also affect the actual runtime of NBSim.

**Effect of $\gamma$:** When $\gamma$ is set to a higher value, the size of the set $C$ returned by QCextract is smaller. As a result, the condition in line 3 of Algorithm 2 is less likely to be satisfied, leading to fewer opportunities to update the variable $S$. Consequently, fewer branches are pruned, requiring more iterations to find the candidate vertex. Thus, the runtime of NBSim may increase when $\gamma$ is set to a higher value.

**Effect of $b$:** In QCextract, the parameter $b$ determines the threshold for the proportion of $|S|$ occupied by $|N(u)|$. A higher value of $b$ results in a stricter condition for considering $S$ as candidate vertices. Consequently, the runtime of NBSim may increase when $b$ is set to a higher value because it could take more iterations to find the near-maximum quasi-clique.

In Section 6, we present an empirical sensitivity analysis of parameters $\gamma$ and $b$ on the accuracy and runtime of the algorithm.

## 5 FASTNBSIM: A MINHASH-BASED ALGORITHM

For real-world large graphs, some vertices can have a very high degree, and their neighbors may need to be iterated repeatedly when computing the containment scores. This can be quite time-consuming with a time complexity of $O(m \cdot d_{max})$. To improve efficiency, we propose approximating the containment score calculations via MinHash signatures.

To efficiently derive approximate similarity scores between adjacent vertex pairs, we adopt the $k$-MinHash technique proposed by Tseng et al. [33]. The key idea is to represent each vertex's neighborhood using a MinHash signature, and then estimate similarity by comparing signatures.

Specifically, we first assign a unique hash value to each vertex $u \in V$. For each $u$, we compute $r_{min}(u)$, the minimum hash value among all vertices in $N(u)$. The Jaccard similarity $\sigma(u, v)$ between vertices $u$ and $v$ can then be given as:

$$\sigma(u, v) = Pr[r_{min}(u) = r_{min}(v)]. \tag{8}$$

To better estimate the probability, i.e., the similarity, we generate $k$ min hashes for each vertex using $k$ independent hash functions. Let $r^i_{min}(u)$ denote the minimum hash value among all vertices in

---

**Algorithm 3:** Extract quasi-clique by MinHash: QCMinHash

**Input:** vertex $u$, threshold $\gamma \in (0, 1]$, $b \in (0, 1]$
**Output:** a vertex set extracted from $N(u)$

1   $S \leftarrow \emptyset$
2   **if** *the MinHash signature of $u$ is not computed* **then**
3     |   Compute the signature of $u$, i.e., $\{r^i_{min}(u) \mid 1 \le i \le k\}$
4   **for** *each vertex $v$ in $N(u)$* **do**
5     |   **if** *the MinHash signature of $v$ is not computed* **then**
6     |    |   Compute the signature of $v$
7     |   Derive the estimated similarity $\hat{\sigma}(u, v)$ with the signatures of $u$ and $v$ via Equation (9)
8     |   Compute $\hat{t}(u, v)$ via Equation (10)
9     |   **if** $\hat{t}(u, v) \ge \gamma$ **then** $S \leftarrow S \cup \{v\}$
10   **if** $\frac{|S|-1}{|N(u)|} < b$ **then** $S \leftarrow \emptyset$
11   **return** $S$

---

$N(u)$ with respect to the $i$-th hash function. We can then estimate $\sigma(u, v)$ as:

$$\hat{\sigma}(u, v) \approx \frac{|\{i \mid r^i_{min}(u) = r^i_{min}(v), 1 \le i \le k\}|}{k}. \tag{9}$$

Next, the estimated Jaccard similarity needs to be converted to the containment score to serve the quasi-clique extraction. Specifically, we introduce a transformation function via the inclusion-exclusion principle to compute the corresponding estimated containment score, $\hat{t}(u, v)$:

$$\hat{t}(u, v) = \frac{(\frac{d(v)+1}{d(u)+1} + 1) * \hat{\sigma}(u, v)}{1 + \hat{\sigma}(u, v)} \tag{10}$$

Combining Equations (9) and (10), we can approximate the containment score for two neighborhoods by MinHash. This process to estimate the containment score is further illustrated in the following example.

*Example 5.1.* Consider the simple graph in Fig. 2. The neighborhood of vertex $u$, denoted $N(u)$, consists of the vertices $\{a, b, c, u, v\}$, while $N(v)$ consists of $\{a, b, d, e, f, u, v\}$. The intersection $N(u) \cap N(v)$ yields $\{a, b, u, v\}$. By Eq. 2, the direct containment score $t(u, v)$ is 0.8.

Now let's approximate $t(u, v)$ using MinHash with $k = 4$ functions: $y = (2x + 3) \mod 11$, $y = (3x + 3) \mod 11$, $y = (2x + 6) \mod 11$, $y = (4x + 4) \mod 11$, and IDs $\{a = 1, b = 2, c = 3, d = 4, e = 5, f = 6, u = 7, v = 8\}$. The signatures are $\{r_{min}(u)\} = \{5, 1, 0, 1\}$ and $\{r_{min}(u)\} = \{0, 2, 0, 1\}$. With $\hat{\sigma}(u, v) = 0.5$ based on Equation (9), the estimated $\hat{t}(u, v) = 0.8$, equal to the direct calculation. □

Based on the above discussion, we propose the MinHash-based quasi-clique extraction algorithm from the neighborhood in Algorithm 3, which follows a similar structure to Algorithm 1 but differs in on-demand signature generation and score estimation, which is shown in the shaded regions of the two algorithms. Specifically, the MinHash signatures are computed on-demand when needed - computing the signature for $u$ if not done yet (lines 2-3), and computing the signature for $v$ if needed (lines 5-6). Then it derives the estimated Jaccard similarity $\hat{\sigma}(u, v)$ and corresponding containment

score $\hat{t}(u, v)$ using the lazily computed signatures and Equations 9 and 10 (lines 7-8).

By computing signatures lazily and estimating scores via Min-Hash, the algorithm aims to efficiently extract quasi-cliques without expensive direct neighborhood comparisons.

**Effect of $k$.** Larger MinHash signature size $k$ leads to a better approximation of the Jaccard similarity. However, bigger $k$ also increases the computation time for signature generation and similarity estimation.

Theoretically, we can give the lower bound of the edge-density of the subgraph returned by Algorithm 3 via the following lemma.

LEMMA 5.2. *Given a graph $G$, a vertex $u$, $k \geq \frac{\ln(nm)}{2\rho^2}$, and threshold parameters $\gamma$ and $b$, the edge-density of the quasi-clique $S$ returned by Algorithm 3 is lower bounded by, if $S$ is not empty:*

$$\delta(S) \geq 1 - \frac{1 - \gamma'}{b} \tag{11}$$

*where*

$$\gamma' \geq \frac{\beta_u \cdot (\beta_u \cdot \rho - \rho \cdot \gamma - \gamma)}{\beta_u \cdot \rho - \rho \cdot \gamma - \beta_u}. \tag{12}$$

*and $\beta_u$ represents $\frac{d_{max}+1}{d(u)+1} + 1$ for the specific vertex $u$.*

PROOF. By setting $k >= \ln(nm)/(2\rho^2)$, we have $\hat{\sigma}(u, v) \in [\sigma(u, v) - \rho, \sigma(u, v) + \rho]$ [33]. Given that $\hat{t}(u, v)$ is required to be larger than $\gamma$, by applying Equation (10), we have

$$\hat{t}(u, v) = \frac{\beta \cdot \hat{\sigma}(u, v)}{1 + \hat{\sigma}(u, v)} \geq \gamma, \tag{13}$$

and

$$t(u, v) = \frac{\beta \cdot \sigma(u, v)}{1 + \sigma(u, v)} \geq \gamma', \tag{14}$$

where $\beta = \frac{d(v)+1}{d(u)+1} + 1$. Combining Equations (13) and (14) and the error bound of $\hat{\sigma}(u, v)$, we can derive:

$$\gamma' \geq \frac{\beta \cdot (\beta \cdot \rho - \rho \cdot \gamma - \gamma)}{\beta \cdot \rho - \rho \cdot \gamma - \beta} \tag{15}$$

Equation (15) exhibits a diminishing trend with increasing values of $\beta$. Since $\beta = \frac{d(v)+1}{d(u)+1} + 1 \leq \frac{d_{max}+1}{d(u)+1} + 1$, by replacing $\beta$ with $\beta_u = \frac{d_{max}+1}{d(u)+1} + 1$, we can derive Equation (12). □

Examining Equation 12, we observe that $\gamma'$ approximates $\gamma$ closely when $\rho$ is set to a small positive real number. This is because large quasi-cliques are typically extracted from high-degree vertices, where $d(u)$ is not very small compared to $d_{max}$. Thus, the degree ratio $\beta_u$ remains low.

Empirically, we find that small $k$ is sufficient for high-quality quasi-clique extraction in many cases.

Algorithm 4 outlines FastNBSim, which modifies NBSim using QCMinHash. The algorithm first constructs a set of $k$ universal hash functions upfront to enable later MinHash computations (line 1). Within the loop, it applies QCMinHash to efficiently extract quasi-cliques from each neighborhood using the MinHash signatures (line 5). The remaining loop order and candidate set updates are identical to the original NBSim in Algorithm 2.

**Complexity.** FastNBSim adapts NBSim to leverage QCMinHash for faster quasi-clique extraction via MinHash approximation. The core steps of NBSim are preserved while substituting direct similarity

---

**Algorithm 4:** Find large quasi-clique: FastNBSim

**Input:** Graph $G$, size $k$, threshold $\gamma$, $b$
**Output:** A near-maximum quasi-clique

1 Construct $k$ universal hash functions
2 $S \leftarrow \emptyset$
3 **for** *each vertex $u$ in descending $\gamma$-degree order* **do**
4     **if** $d_\gamma(u) < T$ **then** break
5     $C \leftarrow$ QCMinHash$(u, \gamma, b)$
6     **if** $|C| > |S|$ **then** $S \leftarrow C$
7 **return** $S$

---

computations with efficient signatures. The time complexity is improved to $O(m \cdot k)$.

## 6 EXPERIMENTS

We now present experimental results. We first discuss the setup in Section 6.1, then describe the results of NBSim and FastNBSim against the baseline algorithms. Then, we give some detailed analysis of the effect of parameters and pruning techniques.

### 6.1 Setup

**Datasets.** We use ten real datasets from [17], and report the number of vertices and edges of each dataset in Table 1. These graphs cover various domains, including co-authorship graphs (e.g., Ca-HepPh and Ca-AstroPh), social networks (e.g., Ego-Facebook and Loc-Gowalla), and web graphs(e.g., Web-Stanford).

**Algorithm**. In our experiments, we employ our newly proposed algorithms NBSim and FastNBSim to compute near-maximum quasi-cliques. For NBSim, we set $\gamma = 0.9$ and $b = 0.6$, FastNBSim follows NBSim with an additional setting $k = 8$. Unless otherwise specified, we use these settings by default. In addition to our algorithms, we also evaluate the performance of the following existing methods:

- NB [16]: This algorithm computes large quasi-cliques using vertex neighborhoods. It can be refined through a straightforward local search method [35], offering state-of-the-art performance with relatively low complexity. The setting of $\alpha$ follows [16].
- NuQClq [7]: As a state-of-the-art algorithm, NuQClq identifies the maximum quasi-clique based on a pre-defined threshold for the quasi-value and a specified cutoff time. The algorithm will terminate when it reaches the cutoff time or the respected result is found. For comparative purposes, we set the quasi threshold to match the quasi-value derived from NBSim and set the cutoff time as sufficiently large to achieve near-optimal results.

All the algorithms above are implemented in C++. For NB, which needs triangle counting, we follow [16] and employ the MAximal Clique Enumerator (MACE) algorithm [36] to obtain triangle counts. We run all the experiments on a machine equipped with an Intel(R) CPU @ 1.4GHz processor and 256GB of memory. The source codes of our algorithms are publicly available [1].

---
[1]https://anonymous.4open.science/r/LargeQCDetection-1DFD/

Table 1: Graphs used in our experiments.

| Dataset | Full name | $|V|$ | $|E|$ |
|---|---|---|---|
| FB | Ego-facebook | 4,039 | 88,234 |
| HP | Ca-HepPh | 12,008 | 118,521 |
| CM | Ca-CondMat | 23,133 | 93,497 |
| ER | Email-Enron | 36,692 | 183,831 |
| GW | Loc-Gowalla | 196,591 | 950,327 |
| SF | Web-Stanford | 281,903 | 2,312,497 |
| BS | Web-BerkStan | 685,230 | 7,600,595 |
| GG | Web-Google | 875,713 | 5,105,039 |
| PK | Soc-Pokec | 1,632,803 | 30,622,564 |
| TC | Wiki-Topcats | 1,791,489 | 28,511,807 |

## 6.2 Main Results

We present the edge-density and size of the quasi-clique returned by each algorithm in Table 2 and Table 3, respectively. We find that in most cases, NBSim and FastNBSim can achieve comparable or even larger sizes with similar edge-density compared with NB and NuQClq.

Table 2: Density of the quasi-clique returned by each method.

| Dataset | NBSim | FastNBSim | NB | NuQClq |
|---|---|---|---|---|
| FB | 0.99 | 0.94 | 0.94 | 0.99 |
| HP | 1 | 1 | 0.95 | 1 |
| CM | 1 | 1 | 0.95 | 1 |
| ER | 0.98 | 0.94 | 0.93 | 0.98 |
| GW | 0.99 | 0.98 | 0.94 | 0.99 |
| SF | 0.99 | 0.94 | 0.95 | 0.99 |
| BS | 0.99 | 0.99 | 0.93 | 0.99 |
| GG | 0.99 | 0.99 | 0.93 | 0.99 |
| PK | 0.98 | 0.98 | 0.95 | 0.98 |
| TC | 0.99 | 0.99 | 0.95 | 0.99 |

Table 3: Size of the quasi-clique returned by each method.

| Dataset | NBSim | FastNBSim | NB | NUQClq |
|---|---|---|---|---|
| FB | 71 | 103 | 50 | 92 |
| HP | 239 | 237 | 246 | 239 |
| CM | 26 | 26 | 28 | 26 |
| ER | 10 | 17 | 14 | 23 |
| GW | 31 | 28 | 36 | 31 |
| SF | 67 | 65 | 71 | 66 |
| BS | 202 | 201 | 142 | 144 |
| GG | 48 | 48 | 54 | 48 |
| PK | 32 | 31 | 33 | 31 |
| TC | 40 | 41 | 48 | 29 |

In Figure 4, we detail the efficiency of all tested algorithms. FastNBSim stands out by markedly enhancing computational efficiency. It achieves speeds up to two orders of magnitude faster

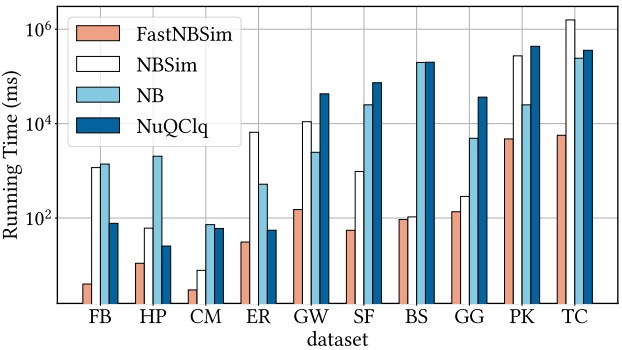

Figure 4: Efficiency of all algorithms.

than NBSim by utilizing MinHash to estimate similarity, thus reducing its time complexity to $O(km)$. Furthermore, when compared to NB and NuQClq, FastNBSim outperforms them, being quicker by up to three orders of magnitude. This pronounced efficiency of FastNBSim can be attributed to its adoption of the MinHash approximation combined with a bound and ordering-based pruning strategy. Conversely, NB necessitates the calculation of the local clustering coefficient for every vertex, leading to a more computationally intensive process.

Turning our attention to NBSim and NB, though they share the same time complexity of $O(m^{3/2})$, their performance varies, each surpassing the other in specific datasets due to different computing paradigms.

## 6.3 Effect of Parameters

*6.3.1 Effect of $\gamma$ and $b$.* From Figure 5, in cases (a), (c), and (e), holding $\gamma$ constant and increasing $b$ shrinks the quasi-clique's size but augments its density for $b$ values between 0.6 and 0.9, aligning with findings in Section 4. Additionally, a rise in $b$ escalates extraction time due to stricter constraints and more candidate clusters, as detailed in Section 4.2. Similarly, cases (b), (d), and (f) illustrate that increasing $\gamma$ with a fixed $b$ mirrors the effects of increasing $b$ with a fixed $r$.

*6.3.2 Effect of varying $k$.* In Figure 6, we report the performance of FastNBSim on datasets HP, ER, GG, and BS varying $k$ from 4 to 128 while fixing $\gamma = 0.9$ and $b = 0.6$. The result of NBSim is marked as "base" in Figure 6 for comparison. Remarkably, HP, GG and BS all exhibit edge densities that are close to 1 for different k values. Overall, both algorithms yield similar and high-quality outcomes. For smaller $k$ values, inaccuracies arise in approximations, yielding larger quasi-clique sizes and decreased edge-densities, especially in ER. Such inaccuracies are attributed to the potential of MinHash to overestimate or underestimate vertex similarities for smaller $k$, as evident in the GG and BS datasets for $k = 4$. However, as $k$ grows, the approximation becomes more accurate.

## 6.4 Effect of Pruning Techniques

*6.4.1 Effect of bound and ordering.* Here we show the effectiveness of the bound and ordering pruning technique proposed in Section 4.2. In Table 4, the proportion of executed branches relative

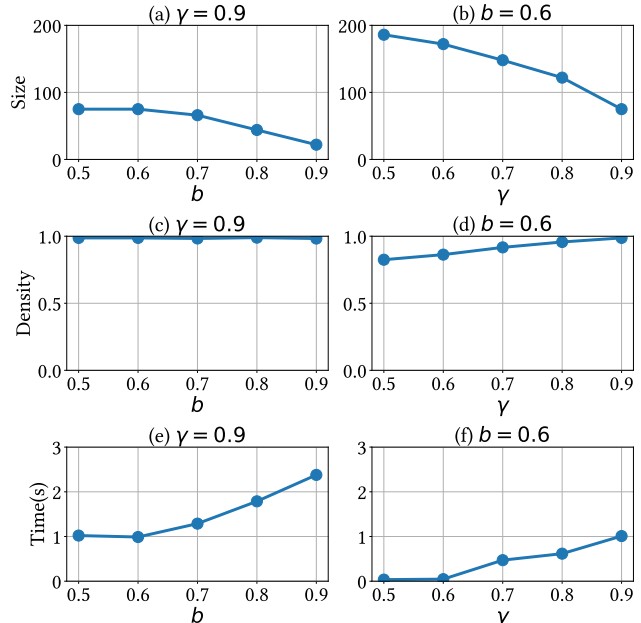

**Figure 5: The accuracy and runtime of** NBSim **on graph FB for different $\gamma$ and $b$. (a), (c), (e) is the result for different $b$ with $\gamma$ fixed to 0.9. (b), (d), (f) is the result for different $\gamma$ with $b$ fixed to 0.6.**

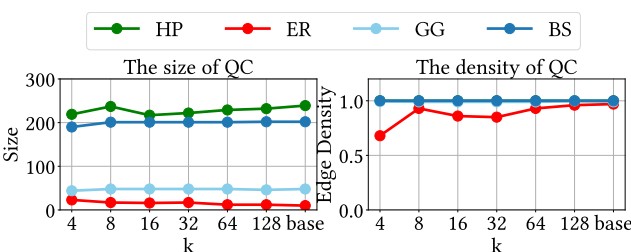

**Figure 6: The quality of QC w.r.t different value of $k$.**

**Table 4: Proportion of neighborhoods examined from the total.**

| Dataset | FB | HP | CM | ER | GW |
|---|---|---|---|---|---|
| Proportion | 2.8% | 0.8% | 0.039% | 6.9% | 2.1% |
| Dataset | SF | BS | GG | PK | TC |
| Proportion | 0.18% | 0.0001% | 0.01% | 12% | 4.9% |

to the total is presented. The total branches equate to the vertex count, indicating that without our pruning strategy, an iteration through every vertex would be necessary. Our findings are drawn from an analysis of ten datasets, all of which consistently exhibit proportion results significantly below 12%. In specific cases, such as CM, GG, and BS, these values are exceptionally low. This underscores the substantial reduction in branches achieved through our pruning approach.

**Table 5: Ratio of signature building time to the overall.**

| Dataset | FB | HP | CM | ER | GW |
|---|---|---|---|---|---|
| Proportion | 63.8% | 49.4% | 15.8% | 71.0% | 68.4% |
| Dataset | SF | BS | GG | PK | TC |
| Proportion | 33.2% | 12.5% | 14.4% | 64.7% | 71.2% |

*6.4.2 Proportion of signature building time.* In Table 5, we showcase the proportions of the signature-building phase as a part of the overall running time for the FastNBSim algorithm across ten datasets when $k = 8$. The high proportions for most datasets underscore that the similarity computation time for FastNBSim is a minimal fraction of the total runtime after signatures are built. In cases where the proportions are relatively low, such as CM, BS, and GG, this is mainly due to the fact that sorting operations occupy the majority of the overall time. For CM, the sort time ratio stands at 79.3%, while for BS, it sits at 75.9%, and for GG, it reaches 84.8%.

**Table 6: Speedup ratio of lazy signature approach compared to calculating all signatures upfront.**

| Dataset | FB | HP | CM | ER | GW |
|---|---|---|---|---|---|
| Speedup | 3.8× | 21.9× | 7.3× | 1.4× | 1.6× |
| Dataset | SF | BS | GG | PK | TC |
| Speedup | 7.9× | 13× | 8.1× | 1.1× | 1.1× |

*6.4.3 Effect of lazy signature construction.* Table 6 presents the speedup ratio of using a lazy signature construction approach versus calculating all signatures upfront for the FastNBSim algorithm. We observe that computing signatures on-demand based on $\gamma$-degree ordering accelerates the runtime since not all vertex signatures need to be computed. On the HP dataset, the speedup ratio is particularly pronounced, demonstrating the efficacy of the proposed lazy signature technique.

# 7 CONCLUSION

In this study, we delved into the maximum quasi-clique problem. We initiated our discussion by reviewing existing algorithms, highlighting their constraints and areas of inefficacy. To enhance the efficiency of the MQC discovery process, we introduced an efficient approximation algorithm, NBSim, and established lower bounds on quasi-clique edge-density. Our efforts further led to the development of an innovative pruning strategy, effectively minimizing redundant computations. Additionally, we integrated an estimation approach for similarity computation using MinHash, culminating in the proposal of the FastNBSim algorithm. This algorithm stands out as it drastically reduces the time complexity associated with similarity score computations to constant time. Through comprehensive experiments on ten real, large-scale datasets, we demonstrated that FastNBSim outpaces existing methods, clocking speeds up to three orders of magnitude faster than state-of-the-art solutions.

In the future, we will explore efficient methods for identifying large quasi-cliques in dynamic graphs, and investigate how to dynamically maintain the MinHash signatures.

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

Received 20 February 2007; revised 12 March 2009; accepted 5 June 2009

