# OpenReview forum: "A Similarity-based Approach for Efficient Large Quasi-clique Detection"
_ACM.org/TheWebConf/2024/Conference — TheWebConf24 Oral_

### Official Review · Reviewer_GQW6 · 2023-11-19

**Novelty:** 5
**Technical Quality:** 6

**Review:**

Computing the maximum quasi-clique is an important problem in graph data analysis. In this paper, the authors propose a similarity-based approach to detect large quasi-clique in graphs. Following the containment score, they propose two algorithms to find near-maximum quasi-cliques by exploiting vertex neighborhood similarity. Extensive experiments are conducted to evaluate the proposed algorithms. However, the proposed algorithms can only find the quasi-clique in the ego-network of a specific vertex, and the proposed algorithms are not evaluated comprehensively, which significantly weakens the contribution of this paper.

Strengths:
S1. The problem studied in the paper is interesting.
S2. New algorithms are proposed to address the problem.
S3. Experiments are conducted to evaluate the proposed algorithms.


Weakness:
W1. The technical contribution of this paper is limited.
W2. The datasets used in the paper is small.
W3. Some part of the algorithm is not well evaluated in the experiments.

**Questions:**

Q1. The proposed method only focuses on the quasi-clique in the ego-network of a specific vertex, which makes the diameter of the detected results is 2 at most all the time. However, based on the definition of quasi-clique, there is no such property, which significantly limited the generalization of the proposed algorithms.

Q2. Based on the problem definition, the given parameter  \alpha also affects the returned results. It is unclear about the performance when directly varying \alpha.

Q3. The datasets used to evaluate the performance is small. Based on the time complexity of the proposed algorithms, it is more convincing if larger datasets could be used to evaluate the performance.

Q4. Some other works related to maximal clique enumeration is not discussed, such as the I/O efficient MCE, diversified clique enumeration.

Q5. Although some guidelines for setting \gamma and b, it is still seems tricky to set  these two parameters appropriately in practice.

**Reviewer Confidence:**

4: The reviewer is certain that the evaluation is correct and very familiar with the relevant literature

**Scope:**

4: The work is relevant to the Web and to the track, and is of broad interest to the community

---

### Official Review · Reviewer_oqGf · 2023-11-21

**Novelty:** 5
**Technical Quality:** 6

**Review:**

The paper considers the quasi-clique finding problem and proposes a new algorithm that utilizes overlapping neighborhood similarities and minhashing. The proposed technique is simple but powerful. The paper is well-written and evaluation is well-done.

- One thing that needs further explanation is how the parameters $\gamma$ and $\beta$ are set. For any new graph, is it always feasible to use the 0.9 and 0.6 values? Is there any connection to the graph structure?

**Questions:**

See above.

**Reviewer Confidence:**

4: The reviewer is certain that the evaluation is correct and very familiar with the relevant literature

**Scope:**

3: The work is somewhat relevant to the Web and to the track, and is of narrow interest to a sub-community

---

### Official Review · Reviewer_jv5T · 2023-11-23

**Novelty:** 5
**Technical Quality:** 5

**Review:**

**Short summary**

The authors address the problem of identifying the maximum quasi clique in an undirected graph. That is, given a graph and a parameter controlling the clique density, the authors develop heuristic algorithms for reporting the largest quasi-clique with density controlled by two parameters (i.e., $\gamma$ and $b$). The authors develop algorithms based on node similarities, i.e., by carefully exploiting node neighborhoods according to node degrees. Additionally, the authors provide an improved variant of their algorithm based on hashing for which they characterize the density of the output of their algorithm under specific conditions. The authors then perform experiments to validate their proposed algorithms, both comparing with state-of-the-art and studying the various parameters of the algorithms.

**Strengths**
1. The problem addressed is interesting and important for the community:
Identifying maximum quasi cliques is an important graph-mining problem with several applications in WWW applications. Additionally, given the hardness of the problem efficient algorithms are required.

2. The paper overall is clearly written and easy to follow: The writing of the paper is simple and easy to follow, most of the techniques proposed by the authors are described such that the reader can understand them besides the technical details.

3. The authors are able to provide some lower bounds, under specific conditions for their output quasi-clique: The obtain some bounds on the density of the quasi-cliques obtained in output to their algorithms. It is unclear if such bounds are tight, and under which condition they can be improved. Perhaps adding such discussion can also help ongoing research in such field. For example, the results obtained in Lemma 5.2, seem not so of practical interest to me (especially if we consider the experimental evaluation, i.e., the values of $k$ used in practice by the authors).

4. The authors provide the source code for their experimental evaluation, a description of public datasets they used, and parameters used for the experimental evaluation.

**Weaknesses**
1. The experimental procedure can be improved.

- To my understanding, the various algorithms compared were executed only once, this can be strengthened by running multiple times the various methods. To attain better statistical power for statistics such as running time.
- Results on memory usage of the various baselines are missing.
- The authors study the impact of the various parameters in their algorithms (such as $\gamma$ and $b$) but they do not provide a hint on how to set them on specific datasets. This can be helpful for someone that needs to execute their algorithms.


2. There could be important missing references (I report some of them below). In particular, I think that a more detailed and exhaustive review is needed, given that the problem has been widely studies, and it is widely related to many other foundational problems.
- Solving maximum quasi-clique problem by a hybrid artificial bee colony approach [Peng et al. (Information Sciences, 2021)]
- Mining Largest Maximal Quasi-Cliques [Sanei-Mehri et al., (TKDD, 2021)]
- On Effectively Finding Maximal Quasi-cliques in Graphs [Brunato et al., (LNCS, 2007)]
- Lightning Fast and Space Efficient k-clique Counting [Ye et al., (WWW, 2022)]
- Provably and Efficiently Approximating Near-cliques using the Turán Shadow: PEANUTS [jain ad Seshadhri, (WWW 2020)]



3. Some aspects of the presentation can be improved.
- Adding a Table showing the different guarantees on the output density, time complexity, as well as memory requirements of the proposed algorithms (also comparing with existing state-of-the-art) can help the overall presentation, showing the achieved improvements.
- In section 6.2 Tables 2 and 3 can be easily merged by using multicolumns, this can save much space.
- Connection to the web should be better highlighted for the proposed problem (e.g., in the introduction), such as finding specific applications that use quasi-cliques for web-based mining tasks.
- The parameter $\rho$ in Lemma 5.2, was never discussed before, and it is not mentioned in such statement.

**Minor and typos**
- Math notation should be properly applied across the manuscript, e.g., line 282 v -> $v$, line 603 >= -> $\ge$, line 799 k -> $k$ (also in Figure 6),  and so on.
-  Line 793: the symbol $r$ was not defined previously.
- In Figure 6, I think the value “base” is not supposed to be there.
- Avoid using symbol "*" to denote the product, just use $\cdot$ or nothing, e.g., line 445, line 553.

**Questions:**

If the authors can argue on the obtained bound on the density of the solution compared to the density of the solution obtained in practice (i.e., how tight the result obtained is in practice).

**Ethics Review Description:**

No issues.

**Reviewer Confidence:**

3: The reviewer is confident but not certain that the evaluation is correct

**Scope:**

3: The work is somewhat relevant to the Web and to the track, and is of narrow interest to a sub-community

---

### Official Review · Reviewer_wNEU · 2023-11-24

**Novelty:** 4
**Technical Quality:** 4

**Review:**

The paper investigates the maximum quasi-clique detection problem and presents an algorithm based on structural similarity in which min-hash is utilized to accelerate the computation. Experiments demonstrate the outperformance on edge density and result size on part of the datasets. Nevertheless, the algorithm design is not quite consistent with the objective of the problem and the theoretical contribution is limited.


Strong points:

S1. The paper adopts structural similarity to refine the SOTA algorithm [16]. It also naturally incorporates min-hash to improve efficiency.

S2. The experimental section validates the effectiveness of all the optimization techniques.

S3. The overall presentation is clear.


Opportunities for improvement:

O1. The objective of the studied problem is to maximize the size of the resulting quasi-clique, while the algorithm design and the experiments emphasize the edge density of the results. If we focus on the objective, the result sizes of the proposed algorithms are smaller than the SOTA algorithms on many datasets, as shown in Table 3. A clarification is needed.

O2. The outperformance of the proposed algorithm is not obvious. In the experiments, the quality of solutions obtained by the NBSim algorithm is similar to NuQClq, but its efficiency is worse on several graphs. The efficiency of FastNBSim is better than NBSim while the result quality is lower. The advantages and the use cases of the proposed algorithms should be made clear.

O3. In comparison to the SOTA solution [16], this paper lacks a rigorous theoretical analysis of the effectiveness of using structural similarity to compute maximum quasi-cliques.

O4. In the review of related works, the shortcomings of prior work in the field and the novelty of the technique design in this paper should be discussed.

O5. It would be better to add explicit connections to the web, e.g. the practical applications of the problem and the case studies on the web data.

**Questions:**

Please refer to O1-O5.

**Reviewer Confidence:**

3: The reviewer is confident but not certain that the evaluation is correct

**Scope:**

3: The work is somewhat relevant to the Web and to the track, and is of narrow interest to a sub-community

---

### Official Review · Reviewer_JuDa · 2023-11-24

**Novelty:** 3
**Technical Quality:** 3

**Review:**

This paper tackles the problem of detecting maximum quasi-cliques.
The main contribution of the paper is to devise methods for this problem which are more efficient and/or effective than the state-of-the-art ones.


The main strengths of the paper are as follows:

S1) The paper comes with convincing motivation.

S2) The proposed methods are well-designed and sound.

S3) The experimental evaluation is well-designed and satisfactorily complete.


The paper also comes with a number of major weaknesses:

W1) The main claims of the paper seem to be not fully supported by experimental evidence. Or, at least, some findings are not sufficiently discussed/motivated. Specifically:
  W1.a) Table 3: NB outperforms NBSim in 8 out of 10 datasets. This somehow contradicts the theoretical design of NBSim, which is mainly devoted to be more effective (and efficient) than NB. The fact that NBSim's quasi-cliques are denser than NB's ones (Table 2) is not a really valid argument, as the two algorithms are supposed and designed to detect maximum-sized cliques, not densest ones.
  W1.b) Figure 4: in several datasets NBSim is (consistently) outperformed by NB. This, again, somehow contradicts the main claims/goal of the paper, according to which NBSim needs to be faster than NB. The authors discuss this by simply stating that it is "due to different computing paradigms". However, at least, a more detailed discussion and precise motivation of this behavior should be provided. In the end, the comparison NBSim vs. NB is central in the paper.
  W1.c) Table 3: FastNBSim detects quasi-cliques of size comparable to or, in two datasets (i.e., FB and ER), consistently larger than the size of the quasi-cliques detected by NBSim. As FastNBSim is an approximation of NBSim, this looks surprising.

W2) The proposed NBSim algorithm (and its faster approximation, FasterNBSim) are not conceptually compared to the state of the art NB algorithm. The main technical differences and novelties, along with the corresponding motivations, should be discussed in detail, otherwise technical contribution and novelty of the proposed method(s) result questionable.

W3) In Section 5, it is said that the time complexity of the proposed NBSim algorithm is O(m d_max), whereas, in Section 6.2, it is said that NBSim shares the same time complexity (O(m^{3/2})) as NB. What is the true time complexity? Also, a detailed time complexity analysis of the proposed NBSim algorithm  (and FastNBSim too) should be provided, as designing more efficient algorithms is a central aspect of the paper.

W4) The paper lacks a proper discussion of how the tackled problem is relevant in a Web setting.

**Questions:**

Please comment on W1.b), W1.c), W2), W3), W4).

**Ethics Review Description:**

No ethical issues.

**Reviewer Confidence:**

3: The reviewer is confident but not certain that the evaluation is correct

**Scope:**

3: The work is somewhat relevant to the Web and to the track, and is of narrow interest to a sub-community

---

### Decision · Program_Chairs · 2024-01-22

**Decision:**

Accept (Oral)

**Comment:**

The reviewers are in concensus about this work's broad relevance to WWW, but there are some concerns about whether the experiments are sufficiently conclusive.